# An Efficient Framework for Clustered Federated Learning

**Avishek Ghosh**[*]
Dept of EECS, UC Berkeley
Berkeley, CA 94720
avishek_ghosh@berkeley.edu

**Jichan Chung**[*]
Dept of EECS, UC Berkeley
Berkeley, CA 94720
jichan3751@berkeley.edu

**Dong Yin**[*]
DeepMind
Mountain View, CA 94043
dongyin@google.com

**Kannan Ramchandran**
Dept of EECS, UC Berkeley
Berkeley, CA 94720
kannanr@berkeley.edu

## Abstract

We address the problem of Federated Learning (FL) where users are distributed and partitioned into clusters. This setup captures settings where different groups of users have their own objectives (learning tasks) but by aggregating their data with others in the same cluster (same learning task), they can leverage the strength in numbers in order to perform more efficient Federated Learning. We propose a new framework dubbed the Iterative Federated Clustering Algorithm (IFCA), which alternately estimates the cluster identities of the users and optimizes model parameters for the user clusters via gradient descent. We analyze the convergence rate of this algorithm first in a linear model with squared loss and then for generic strongly convex and smooth loss functions. We show that in both settings, with good initialization, IFCA converges at an exponential rate, and discuss the optimality of the statistical error rate. When the clustering structure is ambiguous, we propose to train the models by combining IFCA with the weight sharing technique in multi-task learning. In the experiments, we show that our algorithm can succeed even if we relax the requirements on initialization with random initialization and multiple restarts. We also present experimental results showing that our algorithm is efficient in non-convex problems such as neural networks. We demonstrate the benefits of IFCA over the baselines on several clustered FL benchmarks.[2]

## 1 Introduction

In many modern data-intensive applications such as recommendation systems, image recognition, and conversational AI, distributed computing has become a crucial component. In many applications, data are stored in end users' own devices such as mobile phones and personal computers, and in these applications, fully utilizing the on-device machine intelligence is an important direction for next-generation distributed learning. Federated Learning (FL) [28, 17, 27] is a recently proposed distributed computing paradigm that is designed towards this goal, and has received significant attention. Many statistical and computational challenges arise in Federated Learning, due to the highly decentralized system architecture. In this paper, we propose an efficient algorithm that aims to address one of the major challenges in FL—dealing with heterogeneity in the data distribution.

In Federated Learning, since the data source and computing nodes are end users' personal devices, the issue of data heterogeneity, also known as non-i.i.d. data, naturally arises. Exploiting data

---

[*]Equal contribution

[2]Implementation of our experiments is open sourced at https://github.com/jichan3751/ifca.

heterogeneity is particularly crucial in applications such as recommendation systems and personalized advertisement placement, and it benefits both the users' and the enterprises. For example, mobile phone users who read news articles may be interested in different categories of news like politics, sports or fashion; advertisement platforms might need to send different categories of ads to different groups of customers. These indicate that leveraging the heterogeneity among the users is of potential interest—on the one hand, each machine itself may not have enough data and thus we need to better utilize the similarity among the users; on the other hand, if we treat the data from all the users as i.i.d. samples, we may not be able to provide personalized predictions. This problem has recently received much attention [38, 36, 15].

In this paper, we study one of the formulations of FL with non-i.i.d. data, i.e., the *clustered Federated Learning* [36, 26]. We assume that the users are partitioned into different clusters; for example, the clusters may represent groups of users interested in politics, sports, etc, and our goal is to train models for every cluster of users. We note that cluster structure is very common in applications such as recommender systems [35, 23]. The main challenge of our problem is that the *cluster identities of the users are unknown*, and we have to simultaneously solve two problems: identifying the cluster membership of each user and optimizing each of the cluster models in a distributed setting. In order to achieve this goal, we propose a framework and analyze a distributed method, named the *Iterative Federated Clustering Algorithm (IFCA)* for clustered FL. The basic idea of our algorithm is a strategy that alternates between estimating the cluster identities and minimizing the loss functions, and thus can be seen as an Alternating Minimization algorithm in a distributed setting. One of the major advantages of our algorithm is that it does not require a centralized clustering algorithm, and thus significantly reduces the computational cost at the center machine. When the cluster structure is ambiguous, we propose to leverage the weight sharing technique in multi-task learning [3] and combine it with IFCA. More specifically, we learn the shared representation layers using data from all the users, and use IFCA to train separate final layers for each individual cluster.

We further establish convergence rates of our algorithm, for both linear models and general strongly convex losses under the assumption of good initialization. We prove exponential convergence speed, and for both settings, we can obtain *near optimal* statistical error rates in certain regimes. We also present experimental evidence of its performance in practical settings: We show that our algorithm can succeed even if we relax the initialization requirements with random initialization and multiple restarts; and we also present results showing that our algorithm is efficient on neural networks. We demonstrate the effectiveness of IFCA on two clustered FL benchmarks created based on the MNIST and CIFAR-10 datasets, respectively, as well as the Federated EMNIST dataset [2] which is a more realistic benchmark for FL and has ambiguous cluster structure.

Here, we emphasize that clustered Federated Learning is not the only approach to modeling the non-i.i.d. nature of the problem, and different algorithms may be more suitable for different application scenarios; see Section 2 for more discussions. That said, our approach to modeling and the resulting IFCA framework is certainly an important and relatively unexplored direction in Federated Learning. We would also like to note that our theoretical analysis makes contributions to statistical estimation problems with latent variables in distributed settings. In fact, both mixture of regressions [7] and mixture of classifiers [39] can be considered as special cases of our problem in the centralized setting. We discuss more about these algorithms in Section 2.

**Notation:** We use $[r]$ to denote the set of integers $\{1, 2, \ldots, r\}$. We use $\|\cdot\|$ to denote the $\ell_2$ norm of vectors. We use $x \gtrsim y$ if there exists a sufficiently large constant $c > 0$ such that $x \geq cy$, and define $x \lesssim y$ similarly. We use $\mathrm{poly}(m)$ to denote a polynomial in $m$ with arbitrarily large constant degree.

## 2 Related work

During the preparation of the initial draft of this paper, we became aware of a concurrent and independent work by Mansour et al. [26], in which the authors propose clustered FL as one of the formulations for personalization in Federated Learning. The algorithms proposed in our paper and by Mansour et al. are similar. However, our paper makes an important contribution by establishing the *convergence rate* of the *population loss function* under good initialization, which simultaneously guarantees both convergence of the training loss and generalization to test data; whereas in [26], the authors provided only *generalization* guarantees. We discuss other related work in the following.

**Federated Learning and non-i.i.d. data:** Learning with a distributed computing framework has been studied extensively in various settings [50, 32, 22]. As mentioned in Section 1, Federated

Learning [28, 27, 17, 13] is one of the modern distributed learning frameworks that aims to better utilize the data and computing power on edge devices. A central problem in FL is that the data on the users' personal devices are usually non-i.i.d. Several formulations and solutions have been proposed to tackle this problem. A line of research focuses on learning a single global model from non-i.i.d. data [49, 34, 21, 37, 24, 30]. Other lines of research focus more on learning personalized models [38, 36, 8]. In particular, the MOCHA algorithm [38] considers a multi-task learning setting and forms a deterministic optimization problem with the correlation matrix of the users being a regularization term. Our work differs from MOCHA since we consider a statistical setting with cluster structure. Another approach is to formulate Federated Learning with non-i.i.d. data as a meta learning problem [4, 15, 8]. In this setup, the objective is to first obtain a single global model, and then each device fine-tunes the model using its local data. The underlying assumption of this formulation is that the data distributions among different users are similar, and the global model can serve as a good initialization. The formulation of clustered FL has been considered in two recent works [36, 10]. Both of the two works use *centralized* clustering algorithm such as $K$-means, in which the center machine has to identify the cluster identities of all the users, leading to high computational cost at the center. As a result, these algorithms may not be suitable for large models such as deep neural networks or applications with a large number of users. In contrast, our algorithm uses a decentralized approach to identify the cluster identities and thus is more suitable for large-scale applications.

**Latent variable problems:** As mentioned in Section 1, our formulation can be considered as a statistical estimation problem with latent variables in a distributed setting, and the latent variables are the cluster identities. Latent variable problem is a classical topic in statistics and non-convex optimization; examples include Gaussian mixture models (GMM) [44, 20], mixture of linear regressions [7, 43, 48], and phase retrieval [9, 29]. Expectation Maximization (EM) and Alternating Minimization (AM) are two popular approaches to solving these problems. Despite the wide applications, their convergence analyses in the finite sample setting are known to be hard, due to the non-convexity nature of their optimization landscape. In recent years, some progress has been made towards understanding the convergence of EM and AM in the centralized setting [31, 5, 47, 1, 42]. For example, if started from a suitable point, they have fast convergence rate, and occasionally they enjoy super-linear speed of convergence [44, 11]. In this paper, we provide new insights to these algorithms in the FL setting.

## 3 Problem formulation

We begin with a standard statistical learning setting of empirical risk minimization (ERM). Our goal is to learn parametric models by minimizing some loss functions defined by the data. We consider a distributed learning setting where we have one center machine and $m$ worker machines (i.e., each worker machine corresponds to a user in the Federated Learning framework). The center machine and worker machines can communicate with each other using some predefined communication protocol. We assume that there are $k$ different data distributions, $\mathcal{D}_1, \ldots, \mathcal{D}_k$, and that the $m$ machines are partitioned into $k$ disjoint clusters, $S_1^*, \ldots, S_k^*$. We assume no knowledge of the cluster identity of each machine, i.e., the partition $S_1^*, \ldots, S_k^*$ is not revealed to the learning algorithm. We assume that every worker machine $i \in S_j^*$ contains $n$ i.i.d. data points $z^{i,1}, \ldots, z^{i,n}$ drawn from $\mathcal{D}_j$, where each data point $z^{i,j}$ consists of a pair of feature and response denoted by $z^{i,\ell} = (x^{i,\ell}, y^{i,\ell})$.

Let $f(\theta; z) : \Theta \to \mathbb{R}$ be the loss function associated with data point $z$, where $\Theta \subseteq \mathbb{R}^d$ is the parameter space. In this paper, we choose $\Theta = \mathbb{R}^d$. Our goal is to minimize the population loss function $F^j(\theta) := \mathbb{E}_{z \sim \mathcal{D}_j}[f(\theta; z)]$ for all $j \in [k]$. For the purpose of theoretical analysis in Section 5, we focus on the strongly convex losses, in which case we can prove guarantees for estimating the unique solution that minimizes each population loss function. In particular, we try to find solutions $\{\widehat{\theta}_j\}_{j=1}^k$ that are close to $\theta_j^* = \operatorname{argmin}_{\theta \in \Theta} F^j(\theta), j \in [k]$. In our problem, since we only have access to finite data, we take advantage of the empirical loss functions. In particular, let $Z \subseteq \{z^{i,1}, \ldots, z^{i,n}\}$ be a subset of the data points on the $i$-th machine. We define the empirical loss associated with $Z$ as $F_i(\theta; Z) = \frac{1}{|Z|} \sum_{z \in Z} f(\theta; z)$. When it is clear from the context, we may also use the shorthand notation $F_i(\theta)$ to denote an empirical loss associated with some (or all) data on the $i$-th worker.

## 4 Algorithm

In this section, we provide details of our algorithm. We name this scheme *Iterative Federated Clustering Algorithm* (IFCA). The main idea is to alternatively minimize the loss functions while

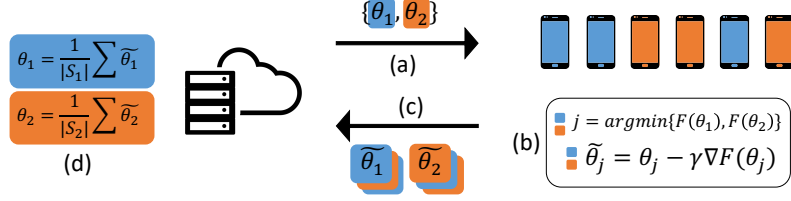

Figure 1: An overview of IFCA (model averaging). (a) The server broadcast models. (b) Worker machines identify their cluster memberships and run local updates. (c) The worker machines send back the local models to server. (d) Average the models within the same estimated cluster $S_j$.

estimating the cluster identities. We discuss two variations of IFCA, namely gradient averaging and model averaging. The algorithm is formally presented in Algorithm 1 and illustrated in Figure 1.

---

**Algorithm 1:** Iterative Federated Clustering Algorithm (IFCA)

---

1: **Input:** number of clusters $k$, step size $\gamma$, $j \in [k]$, initialization $\theta_j^{(0)}$, $j \in [k]$
   number of parallel iterations $T$, number of local gradient steps $\tau$ (for model averaging).
2: **for** $t = 0, 1, \ldots, T - 1$ **do**
3:    center machine: broadcast $\theta_j^{(t)}$, $j \in [k]$
4:    $M_t \leftarrow$ random subset of worker machines (participating devices)
5:    **for** worker machine $i \in M_t$ in parallel **do**
6:       cluster identity estimate $\widehat{j} = \operatorname{argmin}_{j \in [k]} F_i(\theta_j^{(t)})$
7:       define one-hot encoding vector $s_i = \{s_{i,j}\}_{j=1}^{k}$ with $s_{i,j} = \mathbf{1}\{j = \widehat{j}\}$
8:       **option I** (gradient averaging):
9:          compute (stochastic) gradient: $g_i = \widehat{\nabla} F_i(\theta_{\widehat{j}}^{(t)})$, send back $s_i$, $g_i$ to the center machine
10:       **option II** (model averaging):
11:          $\widetilde{\theta}_i = \mathsf{LocalUpdate}(\theta_{\widehat{j}}^{(t)}, \gamma, \tau)$, send back $s_i$, $\widetilde{\theta}_i$ to the center machine
12:    **end for**
13:    center machine:
14:    **option I** (gradient averaging): $\theta_j^{(t+1)} = \theta_j^{(t)} - \frac{\gamma}{m} \sum_{i \in M_t} s_{i,j} g_i$, $\forall\, j \in [k]$
15:    **option II** (model averaging): $\theta_j^{(t+1)} = \sum_{i \in M_t} s_{i,j} \widetilde{\theta}_i / \sum_{i \in M_t} s_{i,j}$, $\forall\, j \in [k]$
16: **end for**
17: **return** $\theta_j^{(T)}$, $j \in [k]$
   $\mathsf{LocalUpdate}(\widetilde{\theta}^{(0)}, \gamma, \tau)$ at the $i$-th worker machine
18: **for** $q = 0, \ldots, \tau - 1$ **do**
19:    (stochastic) gradient descent $\widetilde{\theta}^{(q+1)} = \widetilde{\theta}^{(q)} - \gamma \widehat{\nabla} F_i(\widetilde{\theta}^{(q)})$
20: **end for**
21: **return** $\widetilde{\theta}^{(\tau)}$

---

The algorithm starts with $k$ initial model parameters $\theta_j^{(0)}$, $j \in [k]$. In the $t$-th iteration of IFCA, the center machine selects a random subset of worker machines, $M_t \subseteq [m]$, and broadcasts the current model parameters $\{\theta_j^{(t)}\}_{j=1}^{k}$ to the worker machines in $M_t$. Here, we call $M_t$ the set of *participating devices*. Recall that each worker machine is equipped with local empirical loss function $F_i(\cdot)$. Using the received parameter estimates and $F_i$, the $i$-th worker machine ($i \in M_t$) estimates its cluster identity via finding the model parameter with lowest loss, i.e., $\widehat{j} = \operatorname{argmin}_{j \in [k]} F_i(\theta_j^{(t)})$ (ties can be broken arbitrarily). If we choose the option of gradient averaging, the worker machine then computes a (stochastic) gradient of the local empirical loss $F_i$ at $\theta_{\widehat{j}}^{(t)}$, and sends its cluster identity estimate and gradient back to the center machine. After receiving the gradients and cluster identity estimates from all the participating worker machines, the center machine then collects all the gradient updates from worker machines whose cluster identity estimates are the same and conducts gradient descent update on the model parameter of the corresponding cluster. If we choose the option of model averaging

(similar to the Federated Averaging algorithm [27]), each participating device needs to run $\tau$ steps of local (stochastic) gradient descent updates, get the updated model, and send the new model and its cluster identity estimate to the center machine. The center machine then averages the new models from the worker machines whose cluster identity estimates are the same.

## 4.1 Practical implementation of IFCA

We clarify a few issues regarding the practical implementation of IFCA. In some real-world problems, the cluster structure may be ambiguous, which means that although the distributions of data from different clusters are different, there exists some common properties of the data from all the users that the model should leverage. For these problems, we propose to use the weight sharing technique in multi-task learning [3] and combine it with IFCA. More specifically, when we train neural network models, we can share the weights for the first a few layers among all the clusters so that we can learn a good representation using all the available data, and then run IFCA algorithm only on the last (or last few) layers to address the different distributions among different clusters. Using the notation in Algorithm 1, we run IFCA on a subset of the coordinates of $\theta_j^{(t)}$, and run vanilla gradient averaging or Federated Averaging on the remaining coordinates. Another benefit of this implementation is that we can reduce the communication cost: Instead of sending $k$ models to all the worker machines, the center machine only needs to send $k$ different versions of a subset of all the weights, and one single copy of the shared layers.

Another technique to reduce communication cost is that when the center machine observes that the cluster identities of all the worker machines are stable, i.e., the estimates of their cluster identities do not change for several parallel iterations, then the center machine can stop sending $k$ models to each worker machine, and instead, it can simply send the model corresponding to each worker machine's cluster identity estimate.

## 5 Theoretical guarantees

In this section, we present convergence guarantees of IFCA. In order to streamline our theoretical analysis, we make several simplifications: we consider the IFCA with gradient averaging, and assume that all the worker machines participate in every rounds of IFCA, i.e., $M_t = [m]$ for all $t$. In addition, we also use the *re-sampling* technique for the purpose of theoretical analysis. In particular, suppose that we run a total of $T$ parallel iterations. We partition the $n$ data points on each machine into $2T$ disjoint subsets, each with $n' = \frac{n}{2T}$ data points. For the $i$-th machine, we denote the subsets as $\widehat{Z}_i^{(0)}, \ldots, \widehat{Z}_i^{(T-1)}$ and $Z_i^{(0)}, \ldots, Z_i^{(T-1)}$. In the $t$-th iteration, we use $\widehat{Z}_i^{(t)}$ to estimate the cluster identity, and use $Z_i^{(t)}$ to conduct gradient descent. As we can see, we use fresh data samples for each iteration of the algorithm. Furthermore, in each iteration, we use different set of data points for obtaining the cluster estimate and computing the gradient. This is done in order to remove the inter-dependence between the cluster estimation and the gradient computation, and ensure that in each iteration, we use fresh i.i.d. data that are independent of the current model parameter. We would like to emphasize that re-sampling is a standard tool used in statistics [31, 14, 46, 47, 11], and that it is for theoretical tractability only and is not required in practice as we show in Section 6.

Under these conditions, the update rule for the parameter vector of the $j$-th cluster can be written as

$$S_j^{(t)} = \{i \in [m] : j = \operatorname{argmin}_{j' \in [k]} F_i(\theta_{j'}^{(t)}; \widehat{Z}_i^{(t)})\}, \quad \theta_j^{(t+1)} = \theta_j^{(t)} - \frac{\gamma}{m} \sum_{i \in S_j^{(t)}} \nabla F_i(\theta_j^{(t)}; Z_i^{(t)}),$$

where $S_j^{(t)}$ denotes the set of worker machines whose cluster identity estimate is $j$ in the $t$-th iteration. In the following, we discuss the convergence guarantee of IFCA under two models: in Section 5.1, we analyze the algorithm under a linear model with Gaussian features and squared loss, and in Section 5.2, we analyze the algorithm under a more general setting of strongly convex loss functions.

### 5.1 Linear models with squared loss

In this section, we analyze our algorithm in a concrete linear model. This model can be seen as a warm-up example for more general problems with strongly convex loss functions that we discuss in Section 5.2, as well as a distributed formulation of the widely studied mixture of linear regression problem [46, 47]. We assume that the data on the worker machines in the $j$-th cluster are generated in the following way: for $i \in S_j^*$, the feature-response pair of the $i$-th worker machine machine satisfies

$$y^{i,\ell} = \langle x^{i,\ell}, \theta_j^* \rangle + \epsilon^{i,\ell},$$

where $x^{i,\ell} \sim \mathcal{N}(0, I_d)$ and the additive noise $\epsilon^{i,\ell} \sim \mathcal{N}(0, \sigma^2)$ is independent of $x^{i,\ell}$. Furthermore, we use the squared loss function $f(\theta; x, y) = (y - \langle x, \theta \rangle)^2$. As we can see, this model is the mixture of linear regression model in the distributed setting. We observe that under the above setting, the parameters $\{\theta_j^*\}_{j=1}^k$ are the minimizers of the population loss function $F^j(\cdot)$.

We proceed to analyze our algorithm. We define $p_j := |S_j^*|/m$ as the fraction of worker machines belonging to the $j$-th cluster, and let $p := \min\{p_1, p_2, \ldots, p_k\}$. We also define the minimum separation $\Delta$ as $\Delta := \min_{j \neq j'} \|\theta_j^* - \theta_{j'}^*\|$, and $\rho := \frac{\Delta^2}{\sigma^2}$ as the signal-to-noise ratio. Before we establish our convergence result, we state a few assumptions. Here, recall that $n'$ denotes the number of data that each worker uses in each step.

**Assumption 1.** *The initialization of parameters $\theta_j^{(0)}$ satisfy $\|\theta_j^{(0)} - \theta_j^*\| \leq \frac{1}{4}\Delta, \forall j \in [k]$.*

**Assumption 2.** *Without loss of generality, we assume that $\max_{j \in [k]} \|\theta_j^*\| \lesssim 1$, and that $\sigma \lesssim 1$. We also assume that $n' \gtrsim (\frac{\rho+1}{\rho})^2 \log m$, $d \gtrsim \log m$, $p \gtrsim \frac{\log m}{m}$, $pmn' \gtrsim d$, and $\Delta \gtrsim \frac{\sigma}{p}\sqrt{\frac{d}{mn'}} + \exp(-c(\frac{\rho}{\rho+1})^2 n')$ for some universal constant $c$.*

In Assumption 1, we assume that the initialization is close enough to $\theta_j^*$. We note that this is a standard assumption in the convergence analysis of mixture models [1, 45], due to the non-convex optimization landscape of mixture model problems. In Assumption 2, we put mild assumptions on $n', m, p$, and $d$. The condition that $pmn' \gtrsim d$ simply assumes that the total number of data that we use in each iteration for each cluster is at least as large as the dimension of the parameter space. The condition that $\Delta \gtrsim \frac{\sigma}{p}\sqrt{\frac{d}{mn'}} + \exp(-c(\frac{\rho}{\rho+1})^2 n')$ ensures that the iterates stay close to $\theta_j^*$.

We first provide a single step analysis of our algorithm. We assume that at a certain iteration, we obtain parameter vectors $\theta_j$ that are close to the ground truth parameters $\theta_j^*$, and show that $\theta_j$ converges to $\theta_j^*$ at an exponential rate with an error floor.

**Theorem 1.** *Consider the linear model and assume that Assumptions 1 and 2 hold. Suppose that in a certain iteration of the IFCA algorithm we obtain parameter vectors $\theta_j$ with $\|\theta_j - \theta_j^*\| \leq \frac{1}{4}\Delta$. Let $\theta_j^+$ be iterate after this iteration. Then there exist universal constants $c_1, c_2, c_3, c_4 > 0$ such that when we choose step size $\gamma = c_1/p$, with probability at least $1 - 1/\text{poly}(m)$, we have for all $j \in [k]$,*

$$\|\theta_j^+ - \theta_j^*\| \leq \frac{1}{2}\|\theta_j - \theta_j^*\| + c_2\frac{\sigma}{p}\sqrt{\frac{d}{mn'}} + c_3 \exp\left(-c_4(\frac{\rho}{\rho+1})^2 n'\right).$$

We prove Theorem 1 in Appendix A. Here, we briefly summarize the proof idea. Using the initialization condition, we show that the set $\{S_j\}_{j=1}^k$ has a significant overlap with $\{S_j^*\}_{j=1}^k$. In the overlapped set, we then argue that the gradient step provides a contraction and error floor due to the basic properties of linear regression. We then bound the gradient norm of the miss-classified machines and add them to the error floor. We complete the proof by combining the contributions of properly classified and miss-classified worker machines. We can then iteratively apply Theorem 1 and obtain accuracy of the final solution $\widehat{\theta}_j$ in the following corollary.

**Corollary 1.** *Consider the linear model and assume that Assumptions 1 and 2 hold. By choosing step size $\gamma = c_1/p$, with probability at least $1 - \frac{\log(\Delta/4\varepsilon)}{\text{poly}(m)}$, after $T = \log\frac{\Delta}{4\varepsilon}$ parallel iterations, we have for all $j \in [k]$, $\|\widehat{\theta}_j - \theta_j^*\| \leq \varepsilon$, where $\varepsilon = c_5\frac{\sigma}{p}\sqrt{\frac{d}{mn'}} + c_6 \exp(-c_4(\frac{\rho}{\rho+1})^2 n')$.*

Let us examine the final accuracy. Since the number of data points on each worker machine $n = 2n'T = 2n' \log(\Delta/4\varepsilon)$, we know that for the smallest cluster, there are a total of $2pmn' \log(\Delta/4\varepsilon)$ data points. According to the minimax estimation rate of linear regression [41], we know that *even if we know the ground truth cluster identities*, we cannot obtain an error rate better than $\mathcal{O}(\sigma\sqrt{\frac{d}{pmn' \log(\Delta/4\varepsilon)}})$. Comparing this rate with our statistical accuracy $\varepsilon$, we can see that the first term $\frac{\sigma}{p}\sqrt{\frac{d}{mn'}}$ in $\varepsilon$ is equivalent to the minimax rate up to a logarithmic factor and a dependency on $p$, and the second term in $\varepsilon$ decays exponentially fast in $n'$, and therefore, our final statistical error rate is *near optimal*.

## 5.2 Strongly convex loss functions

In this section, we study a more general scenario where the population loss functions of the $k$ clusters are strongly convex and smooth. In contrast to the previous section, our analysis do not rely on any

specific statistical model, and thus can be applied to more general machine learning problems. We start with reviewing the standard definitions of strongly convex and smooth functions $F : \mathbb{R}^d \mapsto \mathbb{R}$.

**Definition 1.** *$F$ is $\lambda$-strongly convex if $\forall \theta, \theta'$, $F(\theta') \geq F(\theta) + \langle \nabla F(\theta), \theta' - \theta \rangle + \frac{\lambda}{2}\|\theta' - \theta\|^2$.*

**Definition 2.** *$F$ is $L$-smooth if $\forall \theta, \theta'$, $\|\nabla F(\theta) - \nabla F(\theta')\| \leq L\|\theta - \theta'\|$.*

In this section, we assume that the population loss functions $F^j(\theta)$ are strongly convex and smooth.

**Assumption 3.** *The population loss function $F^j(\theta)$ is $\lambda$-strongly convex and $L$-smooth, $\forall j \in [k]$.*

We note that we do not make any convexity or smoothness assumptions on the individual loss function $f(\theta; z)$. Instead, we make the following distributional assumptions on $f(\theta; z)$ and $\nabla f(\theta; z)$.

**Assumption 4.** *For every $\theta$ and every $j \in [k]$, the variance of $f(\theta; z)$ is upper bounded by $\eta^2$, when $z$ is sampled according to $\mathcal{D}_j$, i.e., $\mathbb{E}_{z \sim \mathcal{D}_j}[(f(\theta; z) - F^j(\theta))^2] \leq \eta^2$*

**Assumption 5.** *For every $\theta$ and every $j \in [k]$, the variance of $\nabla f(\theta; z)$ is upper bounded by $v^2$, when $z$ is sampled according to $\mathcal{D}_j$, i.e., $\mathbb{E}_{z \sim \mathcal{D}_j}[\|\nabla f(\theta; z) - \nabla F^j(\theta)\|_2^2] \leq v^2$*

Bounded variance of gradient is very common in analyzing SGD [6]. In this paper we use loss function value to determine cluster identity, so we also need to have a probabilistic assumption on $f(\theta; z)$. We note that bounded variance is a relatively weak assumption on the tail behavior of probability distributions. In addition to the assumptions above, we still use some definitions from Section 5.1, i.e., $\Delta := \min_{j \neq j'} \|\theta_j^* - \theta_{j'}^*\|$, and $p = \min_{j \in [k]} p_j$ with $p_j = |S_j^*|/m$. We make the following assumptions on the initialization, $n'$, $p$, and $\Delta$.

**Assumption 6.** *Without loss of generality, we assume that $\max_{j \in [k]} \|\theta_j^*\| \lesssim 1$. We also assume that $\|\theta_j^{(0)} - \theta_j^*\| \leq \frac{1}{4}\sqrt{\frac{\lambda}{L}}\Delta$, $\forall j \in [k]$, $n' \gtrsim \frac{k\eta^2}{\lambda^2\Delta^4}$, $p \gtrsim \frac{\log(mn')}{m}$, and that $\Delta \geq \widetilde{\mathcal{O}}(\max\{(n')^{-1/5}, m^{-1/6}(n')^{-1/3}\})$.*

Here, for simplicity, the $\widetilde{\mathcal{O}}$ notation omits any logarithmic factors and quantities that do not depend on $m$ and $n'$. As we can see, again we need to assume good initialization, due to the nature of the mixture model, and the assumptions that we impose on $n'$, $p$, and $\Delta$ are relatively mild; in particular, the assumption on $\Delta$ ensures that the iterates stay close to an $\ell_2$ ball around $\theta_j^*$.

**Theorem 2.** *Suppose Assumptions 3-6 hold. Choose step size $\gamma = 1/L$. Then, with probability at least $1 - \delta$, after $T = \frac{8L}{p\lambda} \log\left(\frac{\Delta}{2\varepsilon}\right)$ parallel iterations, we have for all $j \in [k]$, $\|\widehat{\theta}_j - \theta_j^*\| \leq \varepsilon$, where*

$$\varepsilon \lesssim \frac{vkL\log(mn')}{p^{5/2}\lambda^2\delta\sqrt{mn'}} + \frac{\eta^2 L^2 k\log(mn')}{p^2\lambda^4\delta\Delta^4 n'} + \widetilde{\mathcal{O}}(\frac{1}{n'\sqrt{m}}).$$

We prove Theorem 2 in the Appendix B. Similar to Section 5.1, to prove this result, we first prove a per-iteration contraction

$$\|\theta_j^+ - \theta_j^*\| \leq (1 - \frac{p\lambda}{8L})\|\theta_j - \theta_j^*\| + \widetilde{\mathcal{O}}(\frac{1}{\sqrt{mn'}} + \frac{1}{n'} + \frac{1}{n'\sqrt{m}}), \ \forall j \in [k],$$

and then derive the convergence rate. To better interpret the result, we focus on the dependency on $m$ and $n$ and treat other quantities as constants. Then, since $n = 2n'T$, we know that $n$ and $n'$ are of the same scale up to a logarithmic factor. Therefore, the final statistical error rate that we obtain is $\epsilon = \widetilde{\mathcal{O}}(\frac{1}{\sqrt{mn}} + \frac{1}{n})$. As discussed in Section 5.1, $\frac{1}{\sqrt{mn}}$ is the optimal rate even if we know the cluster identities; thus our statistical rate is near optimal in the regime where $n \gtrsim m$. In comparison with the statistical rate in linear models $\widetilde{\mathcal{O}}(\frac{1}{\sqrt{mn}} + \exp(-n))$, we note that the major difference is in the second term. The additional terms of the linear model and the strongly convex case are $\exp(-n)$ and $\frac{1}{n}$, respectively. We note that this is due to different statistical assumptions: in for the linear model, we assume Gaussian noise whereas here we only assume bounded variance.

# 6 Experiments

In this section, we present our experimental results, which not only validate the theoretical claims in Section 5, but also demonstrate that our algorithm can be efficiently applied beyond the regime we discussed in the theory. We emphasize that we *do not* re-sample fresh data points at each iteration. Furthermore, the requirement on the initialization can be relaxed. More specifically, for linear models, we observe that random initialization with a few restarts is sufficient to ensure convergence of Algorithm 1. In our experiments, we also show that our algorithm works efficiently for problems with non-convex loss functions such as neural networks.

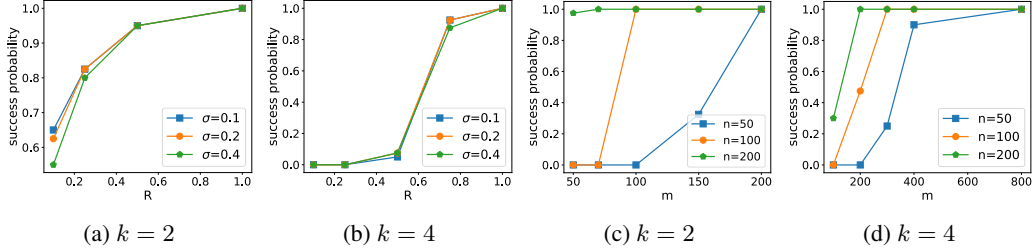

| (a) $k = 2$ | (b) $k = 4$ | (c) $k = 2$ | (d) $k = 4$ |

Figure 2: Success probability with respect to: (a), (b) the separation scale $R$ and the scale of additive noise $\sigma$; (c), (d) the number of worker machines $m$ and the sample size on each machine $n$. In (a) and (b), we see that the success probability gets better with increasing $R$, i.e., more separation between ground truth parameter vectors, and in (c) and (d), we note that the success probability improves with an increase of $mn$, i.e., more data on each machine and/or more machines.

## 6.1 Synthetic data

We begin with evaluation of Algorithm 1 with gradient averaging (option I) on linear models with squared loss, as described in Section 5.1. For all $j \in [k]$, we first generate $\theta_j^* \sim \mathsf{Bernoulli}(0.5)$ coordinate-wise, and then rescale their $\ell_2$ norm to $R$. This ensures that the separation between the $\theta_j^*$'s is proportional to $R$ in expectation, and thus, in this experiment, we use $R$ to represent the *separation* between the ground truth parameter vectors. Moreover, we simulate the scenario where all the worker machines participate in all iterations, and all the clusters contain same number of worker machines. For each trial of the experiment, we first generate the parameter vectors $\theta_j^*$'s, fix them, and then randomly initialize $\theta_j^{(0)}$ according to an independent coordinate-wise Bernoulli distribution. We then run Algorithm 1 for 300 iterations, with a constant step size. For $k = 2$ and $k = 4$, we choose the step size in $\{0.01, 0.1, 1\}$, $\{0.5, 1.0, 2.0\}$, respectively. In order to determine whether we successfully learn the model or not, we sweep over the aforementioned step sizes and define the following distance metric: $\text{dist} = \frac{1}{k} \sum_{j=1}^{k} \|\widehat{\theta}_j - \theta_j^*\|$, where $\{\widehat{\theta}_j\}_{j=1}^{k}$ are the parameter estimates obtained from Algorithm 1. A trial is dubbed *successful* if for a fixed set of $\theta_j^*$, among 10 random initialization of $\theta_j^{(0)}$, at least in one scenario, we obtain $\text{dist} \leq 0.6\sigma$.

In Fig. 2 (a-b), we plot the empirical success probability over 40 trials, with respect to the separation parameter $R$. We set the problem parameters as (a) $(m, n, d) = (100, 100, 1000)$ with $k = 2$, and (b) $(m, n, d) = (400, 100, 1000)$ with $k = 4$. As we can see, when $R$ becomes larger, i.e., the separation between parameters increases, and the problem becomes easier to solve, yielding in a higher success probability. This validates our theoretical result that higher signal-to-noise ratio produces smaller error floor. In Fig. 2 (c-d), we characterize the dependence on $m$ and $n$, with fixing $R$ and $d$ with $(R, d) = (0.1, 1000)$ for (c) and $(R, d) = (0.5, 1000)$ for (d). We observe that when we increase $m$ and/or $n$, the success probability improves. This validates our theoretical finding that more data and/or more worker machines help improve the performance of the algorithm.

## 6.2 Rotated MNIST and CIFAR

We also create clustered FL datasets based on the MNIST [19] and CIFAR-10 [18] datasets. In order to simulate an environment where the data on different worker machines are generated from different distributions, we augment the datasets using rotation, and create the Rotated MNIST [25] and Rotated CIFAR datasets. For **Rotated MNIST**, recall that the MNIST dataset has 60000 training images and 10000 test images with 10 classes. We first augment the dataset by applying $0, 90, 180, 270$ degrees of rotation to the images, resulting in $k = 4$ clusters. For given $m$ and $n$ satisfying $mn = 60000k$, we randomly partition the images into $m$ worker machines so that each machine holds $n$ images *with the same rotation*. We also split the test data into $m_{\text{test}} = 10000k/n$ worker machines in the same way. The **Rotated CIFAR** dataset is also created in a similar way as Rotated MNIST, with the main difference being that we create $k = 2$ clusters with $0$ and $180$ degrees of rotation. We note that creating different tasks by manipulating standard datasets such as MNIST and CIFAR-10 has been widely adopted in the continual learning research community [12, 16, 25]. For clustered FL, creating datasets using rotation helps us simulate a federated learning setup with clear cluster structure.

For our MNIST experiments, we use the fully connected neural network with ReLU activations, with a single hidden layer of size 200; and for our CIFAR experiments, we use a convolution neural network model which consists of 2 convolutional layers followed by 2 fully connected layers, and the images are preprocessed by standard data augmentation such as flipping and random cropping.

Table 1: Test accuracies(%) $\pm$ std on Rotated MNIST ($k = 4$) and Rotated CIFAR ($k = 2$)

| $m, n$ | Rotated MNIST | | | Rotated CIFAR |
|---|---|---|---|---|
| | 4800, 50 | 2400, 100 | 1200, 200 | 200, 500 |
| IFCA (ours) | $\mathbf{94.20 \pm 0.03}$ | $\mathbf{95.05 \pm 0.02}$ | $\mathbf{95.25 \pm 0.40}$ | $\mathbf{81.51 \pm 1.37}$ |
| global model | $86.74 \pm 0.04$ | $88.65 \pm 0.08$ | $89.73 \pm 0.13$ | $77.87 \pm 0.39$ |
| local model | $63.32 \pm 0.02$ | $73.66 \pm 0.04$ | $80.05 \pm 0.02$ | $33.97 \pm 1.19$ |

We compare our IFCA algorithm with two baseline algorithms, i.e., the *global model*, and *local model* schemes. For **IFCA**, we use model averaging (option II in Algorithm 1). For MNIST experiments, we use full worker machines participation ($M_t = [m]$ for all $t$). For LocalUpdate step in Algorithm 1, we choose $\tau = 10$ and step size $\gamma = 0.1$. For CIFAR experiments, we choose $|M_t| = 0.1m$, and apply step size decay 0.99, and we also set $\tau = 5$ and batch size 50 for LocalUpdate process, following prior works [28]. In the **global model** scheme, the algorithm tries to learn single global model that can make predictions from all the distributions. The algorithm does not consider cluster identities, so model averaging step in Algorithm 1 becomes $\theta^{(t+1)} = \sum_{i \in M_t} \widetilde{\theta}_i / |M_t|$, i.e. averaged over parameters from all the participating machines. In the **local model** scheme, the model in each node performs gradient descent only on local data available, and model averaging is not performed.

For IFCA and the global model scheme, we perform inference in the following way. For every test worker machine, we run inference on all learned models ($k$ models for IFCA and one model for global model scheme), and calculate the accuracy from the model that produces the smallest loss value. For testing the local model baselines, the models are tested by measuring the accuracy on the test data with the same distribution (i.e. those have the same rotation). We report the accuracy averaged over all the models in worker machines. For all algorithms, we run experiment with 5 different random seeds and report the average and standard deviation.

Our experimental results are shown in Table 1. We can observe that our algorithm performs better than the two baselines. As we run the IFCA algorithm, we observe that we can gradually find the underlying cluster identities of the worker machines, and after the correct cluster is found, each model is trained and tested using data with the same distribution, resulting in better accuracy. The global model baseline performs worse than ours since it tries to fit all the data from different distributions, and cannot provide personalized predictions. The local model baseline algorithm overfits to the local data easily, leading to worse performance than ours.

## 6.3 Federated EMNIST

We provide additional experimental results on the Federated EMNIST (FEMNIST) [2], which is a realistic FL dataset where the data points on every worker machine are the handwritten digits or letters from a specific writer. Although the data distribution among all the users are similar, there might be ambiguous cluster structure since the writing styles of different people may be clustered. We use the weight sharing technique mentioned in Section 4.1. We use a neural network with two convolutional layers, with a max pooling layer after each convolutional layer, followed by two fully connected layers. We share the weights of all the layers, except the last layer which is trained by IFCA. We treat the number of clusters $k$ as a hyper parameter and run the experiments with different values of $k$. We compare IFCA with the global model and local model approaches, as well as the one-shot centralized clustering algorithm in [10]. The test accuracies are shown in Table 2, with mean and standard deviation computed over 5 independent runs. As we can see, IFCA shows clear advantage over the global model and local model approaches. The results of IFCA and the one-shot algorithm are similar. However, as we emphasized in Section 2, IFCA does not run a centralized clustering procedure, and thus reduces the computational cost at the center machine. As a final note, we observe that IFCA is robust to the choice of the number of clusters $k$. The results of the algorithm with $k = 2$ and $k = 3$ are similar, and we notice that when $k > 3$, IFCA automatically identifies 3 clusters, and the remaining clusters are empty. This indicates the applicability of IFCA in real-world problems where the cluster structure is ambiguous and the number of clusters is unknown.

Table 2: Test accuracies (%) $\pm$ std on FEMNIST

| IFCA ($k = 2$) | IFCA ($k = 3$) | one-shot ($k = 2$) | one-shot ($k = 3$) | global | local |
|---|---|---|---|---|---|
| $87.99 \pm 0.35$ | $87.89 \pm 0.52$ | $87.41 \pm 0.39$ | $87.38 \pm 0.37$ | $84.45 \pm 0.51$ | $75.85 \pm 0.72$ |

## Broader Impact

In this paper, we study the problem of clustered Federated Learning. Our formulation is one of the problem setups for personalized Federated Learning. We expect that, overall our framework will better protect the users' privacy in a Federated Learning system while still provide personalized predictions. The reason is that our algorithm does not require the users to send any of their own personal data to the central server, and the users can still learn a personalized model using their on-device computing power. One potential risk is that our algorithm still requires the users to send the estimates of their cluster identities to the central server. Thus there might still be privacy concerns in this step. We suggest that before applying our algorithm, or generally any FL algorithms, in a real-world system, we should first request the users' consent.

## Acknowledgments and Disclosure of Funding

The authors would like to thank Mehrdad Farajtabar and anonymous reviewers for their helpful comments. This work was partially supported by NSF CIF-1703678 and MLWiNS 2002821. Part of this work was done when Dong Yin was a PhD student at UC Berkeley.

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
