[Supplementary Material]

# Appendix

In our proofs, we use $c, c_1, c_2, \ldots$ to denote positive universal constants, the value of which may differ across instances. For a matrix $A$, we write $\|A\|_{op}$ and $\|A\|_F$ as the operator norm and Frobenius norm, respectively. For a set $S$, we use $\overline{S}$ to denote the complement of the set.

## A  Proof of Theorem 1

Since we only analyze a single iteration, for simplicity we drop the superscript that indicates the iteration counter. Suppose that at a particular iteration, we have model parameters $\theta_j$, $j \in [k]$, for the $k$ clusters. We denote the *estimation* of the set of worker machines that belongs to the $j$-th cluster by $S_j$, and recall that the true clusters are denoted by $S_j^*$, $j \in [k]$.

**Probability of erroneous cluster identity estimation**  We begin with the analysis of the probability of incorrect cluster identity estimation. Suppose that a worker machine $i$ belongs to $S_j^*$. We define the event $\mathcal{E}_i^{j,j'}$ as the event when the $i$-th machine is classified to the $j'$-th cluster, i.e., $i \in S_{j'}$. Thus the event that worker $i$ is correctly classified is $\mathcal{E}_i^{j,j}$, and we use the shorthand notation $\mathcal{E}_i := \mathcal{E}_i^{j,j}$. We now provide the following lemma that bounds the probability of $\mathcal{E}_i^{j,j'}$ for $j' \neq j$.

**Lemma 1.** *Suppose that worker machine $i \in S_j^*$. Let $\rho := \frac{\Delta^2}{\sigma^2}$. Then there exist universal constants $c_1$ and $c_2$ such that for any $j' \neq j$,*

$$\mathbb{P}(\mathcal{E}_i^{j,j'}) \leq c_1 \exp\left(-c_2 n'(\frac{\rho}{\rho+1})^2\right),$$

*and by union bound*

$$\mathbb{P}(\overline{\mathcal{E}_i}) \leq c_1 k \exp\left(-c_2 n'(\frac{\rho}{\rho+1})^2\right).$$

We prove Lemma 1 in Appendix A.1.

Now we proceed to analyze the gradient descent step. Without loss of generality, we only analyze the first cluster. The update rule of $\theta_1$ in this iteration can be written as

$$\theta_1^+ = \theta_1 - \frac{\gamma}{m} \sum_{i \in S_1} \nabla F_i(\theta_1; Z_i),$$

where $Z_i$ is the set of the $n'$ data points that we use to compute gradient in this iteration on a particular worker machine.

We use the shorthand notation $F_i(\theta) := F_i(\theta; Z_i)$, and note that $F_i(\theta)$ can be written in the matrix form as

$$F_i(\theta) = \frac{1}{n'}\|Y_i - X_i\theta\|^2,$$

where we have the feature matrix $X_i \in \mathbb{R}^{n' \times d}$ and response vector $Y_i = X_i\theta_1^* + \epsilon_i$. According to our model, all the entries of $X_i$ are i.i.d. sampled according to $\mathcal{N}(0, 1)$, and $\epsilon_i \sim \mathcal{N}(0, \sigma^2 I)$.

We first notice that

$$\|\theta_1^+ - \theta_1^*\| = \| \underbrace{\theta_1 - \theta_1^* - \frac{\gamma}{m} \sum_{i \in S_1 \cap S_1^*} \nabla F_i(\theta_1)}_{T_1} - \underbrace{\frac{\gamma}{m} \sum_{i \in S_1 \cap \overline{S_1^*}} \nabla F_i(\theta_1)}_{T_2} \| \leq \|T_1\| + \|T_2\|.$$

We control the two terms separately. Let us first focus on $\|T_1\|$.

**Bound $\|T_1\|$**  To simplify notation, we concatenate all the feature matrices and response vectors of all the worker machines in $S_1 \cap S_1^*$ and get the new feature matrix $X \in \mathbb{R}^{N \times d}$, $Y \in \mathbb{R}^N$ with

$Y = X\theta_1^* + \epsilon$, where $N := n'|S_1 \cap S_1^*|$. It is then easy to verify that

$$T_1 = (I - \frac{2\gamma}{mn'}X^\top X)(\theta_1 - \theta_1^*) + \frac{2\gamma}{mn'}X^\top \epsilon$$
$$= (I - \frac{2\gamma}{mn'}\mathbb{E}[X^\top X])(\theta_1 - \theta_1^*) + \frac{2\gamma}{mn'}(\mathbb{E}[X^\top X] - X^\top X)(\theta_1 - \theta_1^*) + \frac{2\gamma}{mn'}X^\top \epsilon$$
$$= (1 - \frac{2\gamma N}{mn'})(\theta_1 - \theta_1^*) + \frac{2\gamma}{mn'}(\mathbb{E}[X^\top X] - X^\top X)(\theta_1 - \theta_1^*) + \frac{2\gamma}{mn'}X^\top \epsilon.$$

Therefore

$$\|T_1\| \leq (1 - \frac{2\gamma N}{mn'})\|\theta_1 - \theta_1^*\| + \frac{2\gamma}{mn'}\|X^\top X - \mathbb{E}[X^\top X]\|_{op}\|\theta_1 - \theta_1^*\| + \frac{2\gamma}{mn'}\|X^\top \epsilon\|. \quad (1)$$

Thus in order to bound $\|T_1\|$, we need to analyze two terms, $\|X^\top X - \mathbb{E}[X^\top X]\|_{op}$ and $\|X^\top \epsilon\|$. To bound $\|X^\top X - \mathbb{E}[X^\top X]\|_{op}$, we first provide an analysis of $N$ showing that it is large enough. Using Lemma 1 in conjunction with Assumption 2, we see that the probability of correctly classifying any worker machine $i$, given by $\mathbb{P}(\mathcal{E}_i)$, satisfies $\mathbb{P}(\mathcal{E}_i) \geq \frac{1}{2}$. Hence, we obtain

$$\mathbb{E}[|S_1 \cap S_1^*|] \geq \mathbb{E}[\frac{1}{2}|S_1^*|] = \frac{1}{2}p_1 m,$$

where we use the fact that $|S_1^*| = p_1 m$. Since $|S_1 \cap S_1^*|$ is a sum of Bernoulli random variables with success probability at least $\frac{1}{2}$, we obtain

$$\mathbb{P}\left(|S_1 \cap S_1^*| \leq \frac{1}{4}p_1 m\right) \leq \mathbb{P}\left(\left||S_1 \cap S_1^*| - \mathbb{E}[|S_1 \cap S_1^*|]\right| \geq \frac{1}{4}p_1 m\right) \leq 2\exp(-cpm),$$

where $p = \min\{p_1, p_2, \ldots, p_k\}$, and the second step follows from Hoeffding's inequality. Hence, we obtain $|S_1 \cap S_1^*| \geq \frac{1}{4}p_1 m$ with high probability, which yields

$$\mathbb{P}(N \geq \frac{1}{4}p_1 mn') \geq 1 - 2\exp(-cpm). \quad (2)$$

By combining this fact with our assumption that $pmn' \gtrsim d$, we know that $N \gtrsim d$. Then, according to the concentration of the covariance of Gaussian random vectors [41], we know that with probability at least $1 - 2\exp(-\frac{1}{2}d)$,

$$\|X^\top X - \mathbb{E}[X^\top X]\|_{op} \leq 6\sqrt{dN} \lesssim N. \quad (3)$$

We now proceed to bound $\|X^\top \epsilon\|$. In particular, we use the following lemma.

**Lemma 2.** *Consider a random matrix $X \in \mathbb{R}^{N \times d}$ with i.i.d. entries sampled according to $\mathcal{N}(0, 1)$, and $\epsilon \in \mathbb{R}^N$ be a random vector sampled according to $\mathcal{N}(0, \sigma^2 I)$, independently of $X$. Then we have with probability at least $1 - 2\exp(-c_1 \max\{d, N\})$,*

$$\|X\|_{op} \leq c \max\{\sqrt{d}, \sqrt{N}\},$$

*and with probability at least $1 - c_2 \exp(-c_3 \min\{d, N\})$,*

$$\|X^\top \epsilon\| \leq c_4 \sigma \sqrt{dN}.$$

We prove Lemma 2 in Appendix A.2. Now we can combine (1), (3), (2), and Lemma 2 and obtain with probability at least $1 - c_1 \exp(-c_2 pm) - c_3 \exp(-c_4 d)$,

$$\|T_1\| \leq (1 - c_5 \gamma p)\|\theta_1 - \theta_1^*\| + c_6 \gamma \sigma \sqrt{\frac{d}{mn'}}. \quad (4)$$

Since we assume that $p \gtrsim \frac{\log m}{m}$ and $d \gtrsim \log m$, the success probability can be simplified as $1 - 1/\mathrm{poly}(m)$.

**Bound $\|T_2\|$** We first condition on $S_1$. We have the following:

$$\nabla F_i(\theta_1) = \frac{2}{n'}X_i^\top(Y_i - X_i\theta_1).$$

For $i \in S_1 \cap S_j^*$, with $j \neq 1$, we have $Y_i = X_i\theta_j^* + \epsilon_i$, and so we obtain

$$n'\nabla F_i(\theta_1) = 2X_i^\top X_i(\theta_j^* - \theta_1) + 2X_i^\top\epsilon_i,$$

which yields

$$n'\|\nabla F_i(\theta_1)\| \lesssim \|X_i\|_{op}^2 + \|X_i^\top\epsilon_i\|, \tag{5}$$

where we use the fact that $\|\theta_j^* - \theta_1\| \leq \|\theta_j^*\| + \|\theta_1^*\| + \|\theta_1^* - \theta_1\| \lesssim 1$. Then, we combine (5) and Lemma 2 and get with probability at least $1 - c_1\exp(-c_2\min\{d, n'\})$,

$$\|\nabla F_i(\theta_1)\| \leq \frac{1}{n'}(c_3\max\{d, n'\} + c_4\sigma\sqrt{dn'}) \leq c_5\max\{1, \frac{d}{n'}\}, \tag{6}$$

where we use our assumption that $\sigma \lesssim 1$. By union bound, we know that with probability at least $1 - c_1 m\exp(-c_2\min\{d, n'\})$, (6) holds for all $j \in \overline{S_1^*}$. In addition, since we assume that $n' \gtrsim \log m$, $d \gtrsim \log m$, this probability can be lower bounded by $1 - 1/\mathrm{poly}(m)$. This implies that conditioned on $S_1$, with probability at least $1 - 1/\mathrm{poly}(m)$,

$$\|T_2\| \leq c_5\frac{\gamma}{m}|S_1 \cap \overline{S_1^*}|\max\{1, \frac{d}{n'}\}. \tag{7}$$

Since we choose $\gamma = \frac{c}{p}$, we have $\frac{\gamma}{m}\max\{1, \frac{d}{n'}\} \lesssim 1$, where we use our assumption that $pmn' \gtrsim d$. This shows that with probability at least $1 - 1/\mathrm{poly}(m)$,

$$\|T_2\| \leq c_5|S_1 \cap \overline{S_1^*}|. \tag{8}$$

We then analyze $|S_1 \cap \overline{S_1^*}|$. By Lemma 1, we have

$$\mathbb{E}[|S_1 \cap \overline{S_1^*}|] \leq c_6 m\exp(-c_7(\frac{\rho}{\rho+1})^2 n'). \tag{9}$$

According to Assumption 2, we know that $n' \geq c(\frac{\rho+1}{\rho})^2\log m$, for some constant $c$ that is large enough. Therefore, $m \leq \exp(\frac{1}{c}(\frac{\rho}{\rho+1})^2 n')$, and thus, as long as $c$ is large enough such that $\frac{1}{c} < c_7$ where $c_7$ is defined in (9), we have

$$\mathbb{E}[|S_1 \cap \overline{S_1^*}|] \leq c_6\exp(-c_8(\frac{\rho}{\rho+1})^2 n'). \tag{10}$$

and then by Markov's inequality, we have

$$\mathbb{P}\left(|S_1 \cap \overline{S_1^*}| \leq c_6\exp(-\frac{c_8}{2}(\frac{\rho}{\rho+1})^2 n')\right) \geq 1 - \exp(-\frac{c_8}{2}(\frac{\rho}{\rho+1})^2 n')) \geq 1 - \mathrm{poly}(m). \tag{11}$$

Combining (8) with (11), we know that with probability at least $1 - 1/\mathrm{poly}(m)$,

$$\|T_2\| \leq c_1\exp(-c_2(\frac{\rho}{\rho+1})^2 n').$$

Using this fact and (4), we obtain that with probability at least $1 - 1/\mathrm{poly}(m)$,

$$\|\theta_1^+ - \theta_1^*\| \leq (1 - c_1\gamma p)\|\theta_1 - \theta_1^*\| + c_2\gamma\sigma\sqrt{\frac{d}{mn'}} + c_3\exp(-c_4(\frac{\rho}{\rho+1})^2 n').$$

Then we can complete the proof for the first cluster by choosing $\gamma = \frac{1}{2c_1 p}$. To complete the proof for all the $k$ clusters, we can use union bound, and the success probability is $1 - k/\mathrm{poly}(m)$. However, since $k \leq m$ by definition, we still have success probability $1 - 1/\mathrm{poly}(m)$.

## A.1 Proof of Lemma 1

Without loss of generality, we analyze $\mathcal{E}_i^{1,j}$ for some $j \neq 1$. By definition, we have

$$\mathcal{E}_i^{1,j} = \{F_i(\theta_j; \widehat{Z}_i) \leq F_i(\theta_1; \widehat{Z}_i)\},$$

where $\widehat{Z}_i$ is the set of $n'$ data points that we use to estimate the cluster identity in this iteration. We write the data points in $\widehat{Z}_i$ in matrix form with feature matrix $X_i \in \mathbb{R}^{n' \times d}$ and response vector $Y_i = X_i \theta_1^* + \epsilon_i$. According to our model, all the entries of $X_i$ are i.i.d. sampled according to $\mathcal{N}(0, 1)$, and $\epsilon_i \sim \mathcal{N}(0, \sigma^2 I)$. Then, we have

$$\mathbb{P}\{\mathcal{E}_i^{1,j}\} = \mathbb{P}\left\{\|X_i(\theta_1^* - \theta_1) + \epsilon_i\|^2 \geq \|X_i(\theta_1^* - \theta_j) + \epsilon_i\|^2\right\}.$$

Consider the random vector $X_i(\theta_1^* - \theta_j) + \epsilon_i$, and in particular consider the $\ell$-th coordinate of it. Since $X_i$ and $\epsilon_i$ are independent and we resample $(X_i, Y_i)$ at each iteration, the $\ell$-th coordinate of $X_i(\theta_1^* - \theta_j) + \epsilon_i$ is a Gaussian random variable with mean 0 and variance $\|\theta_j - \theta_1^*\|^2 + \sigma^2$. Since $X_i$ and $\epsilon_i$ contain independent rows, the distribution of $\|X_i(\theta_1^* - \theta_j) + \epsilon_i\|^2$ is given by $(\|\theta_j - \theta_1^*\|^2 + \sigma^2)u_j$, where $u_j$ is a standard Chi-squared random variable $n'$ degrees of freedom. We now calculate the an upper bound on the following probability:

$$\mathbb{P}\left\{\|X_i(\theta_1^* - \theta_1) + \epsilon_i\|^2 \geq \|X_i(\theta_1^* - \theta_j) + \epsilon_i\|^2\right\}$$
$$\overset{(i)}{\leq} \mathbb{P}\left\{\|X_i(\theta_1^* - \theta_j) + \epsilon_i\|^2 \leq t\right\} + \mathbb{P}\left\{\|X_i(\theta_1^* - \theta_1) + \epsilon_i\|^2 > t\right\}$$
$$\leq \mathbb{P}\left\{(\|\theta_j - \theta_1^*\|^2 + \sigma^2)u_j \leq t\right\} + \mathbb{P}\left\{(\|\theta_1 - \theta_1^*\|^2 + \sigma^2)u_1 > t\right\}, \tag{12}$$

where (i) holds for all $t \geq 0$. For the first term, we use the concentration property of Chi-squared random variables. Using the fact that $\|\theta_j - \theta_1^*\| \geq \|\theta_j^* - \theta_1^*\| - \|\theta_j - \theta_j^*\| \geq \frac{3}{4}\Delta$, we have

$$\mathbb{P}\left\{(\|\theta_j - \theta_1^*\|^2 + \sigma^2)u_j \leq t\right\} \leq \mathbb{P}\left\{(\frac{9}{16}\Delta^2 + \sigma^2)u_j \leq t\right\}. \tag{13}$$

Similarly, using the initialization condition, $\|\theta_1 - \theta_1^*\| \leq \frac{1}{4}\Delta$, the second term of equation (12) can be simplified as

$$\mathbb{P}\left\{(\|\theta_1 - \theta_1^*\|^2 + \sigma^2)u_1 > t\right\} \leq \mathbb{P}\left\{(\frac{1}{16}\Delta^2 + \sigma^2)u_1 > t\right\}. \tag{14}$$

Based on the above observation, we now choose $t = n'(\frac{5}{16}\Delta^2 + \sigma^2)$. Recall that $\rho := \frac{\Delta^2}{\sigma^2}$. Then the inequlity (13) can be rewritten as

$$\mathbb{P}\left\{(\|\theta_j - \theta_1^*\|^2 + \sigma^2)u_j \leq t\right\} \leq \mathbb{P}\left\{\frac{u_j}{n'} - 1 \leq -\frac{4\rho}{9\rho + 16}\right\}.$$

According to the concentration results for standard Chi-squared distribution [41], we know that there exists universal constants $c_1$ and $c_2$ such that

$$\mathbb{P}\left\{(\|\theta_j - \theta_1^*\|^2 + \sigma^2)u_j \leq t\right\} \leq c_1 \exp\left(-c_2 n'(\frac{\rho}{\rho + 1})^2\right). \tag{15}$$

Similarly, the inequality (14) can be rewritten as

$$\mathbb{P}\left\{(\|\theta_1 - \theta_1^*\|^2 + \sigma^2)u_1 > t\right\} \leq \mathbb{P}\left\{\frac{u_1}{n'} - 1 > \frac{4\rho}{\rho + 16},\right\}$$

and again according to the concentration of Chi-squared distribution, there exists universal constants $c_3$ and $c_4$ such that

$$\mathbb{P}\left\{(\|\theta_1 - \theta_1^*\|^2 + \sigma^2)u_1 > t\right\} \leq c_3 \exp\left(-c_4 n'(\frac{\rho}{\rho + 1})^2\right). \tag{16}$$

The proof can be completed by combining (12), (15) and (16).

## A.2 Proof of Lemma 2

According to Theorem 5.39 of [40], we have with probability at least $1 - 2\exp(-c_1 \max\{d, N\})$,

$$\|X\|_{op} \leq c\max\{\sqrt{d}, \sqrt{N}\},$$

where $c$ and $c_1$ are universal constants. As for $\|X^\top \epsilon\|$, we first condition on $X$. According to the Hanson-Wright inequality [33], we obtain for every $t \geq 0$

$$\mathbb{P}\left(\left|\|X^\top \epsilon\| - \sigma\|X^\top\|_F\right| > t\right) \leq 2\exp\left(-c\frac{t^2}{\sigma^2\|X^\top\|_{op}^2}\right). \tag{17}$$

Using Chi-squared concentration [41], we obtain with probability at least $1 - 2\exp(-cdN)$,

$$\|X\|_F \leq c\sqrt{dN}.$$

Furthermore, using the fact that $\|X^\top\|_{op} = \|X\|_{op}$ and substituting $t = \sigma\sqrt{dN}$ in (17), we obtain with probability at least $1 - c_2\exp(-c_3\min\{d, N\})$,

$$\|X^\top \epsilon\| \leq c_4\sigma\sqrt{dN}.$$

# B Proof of Theorem 2

The proof of this theorem is similar to that of the linear model. We begin with a single-step analysis.

## B.1 Analysis for a single step

Suppose that at a certain step, we have model parameters $\theta_j$, $j \in [k]$ for the $k$ clusters. Assume that $\|\theta_j - \theta_j^*\| \leq \frac{1}{4}\sqrt{\frac{\lambda}{L}}\Delta$, for all $j \in [k]$.

**Probability of erroneous cluster identity estimation:** We first calculate the probability of erroneous estimation of worker machines' cluster identity. We define the events $\mathcal{E}_i^{j,j'}$ in the same way as in Appendix A, and have the following lemma.

**Lemma 3.** *Suppose that worker machine $i \in S_j^*$. Then there exists a universal constants $c_1$ such that for any $j' \neq j$,*

$$\mathbb{P}(\mathcal{E}_i^{j,j'}) \leq c_1 \frac{\eta^2}{\lambda^2 \Delta^4 n'},$$

*and by union bound*

$$\mathbb{P}(\overline{\mathcal{E}_i}) \leq c_1 \frac{k\eta^2}{\lambda^2 \Delta^4 n'}.$$

We prove Lemma 3 in Appendix B.3. Now we proceed to analyze the gradient descent iteration. Without loss of generality, we focus on $\theta_1$. We have

$$\|\theta_1^+ - \theta_1^*\| = \|\theta_1 - \theta_1^* - \frac{\gamma}{m}\sum_{i \in S_1}\nabla F_i(\theta_1)\|,$$

where $F_i(\theta) := F_i(\theta; Z_i)$ with $Z_i$ being the set of data points on the $i$-th worker machine that we use to compute the gradient, and $S_1$ is the set of indices returned by Algorithm 1 corresponding to the first cluster. Since

$$S_1 = (S_1 \cap S_1^*) \cup (S_1 \cap \overline{S_1^*})$$

and the sets are disjoint, we have

$$\|\theta_1^+ - \theta_1^*\| = \|\theta_1 - \theta_1^* - \frac{\gamma}{m}\underbrace{\sum_{i \in S_1 \cap S_1^*}\nabla F_i(\theta_1)}_{T_1} - \frac{\gamma}{m}\underbrace{\sum_{i \in S_1 \cap \overline{S_1^*}}\nabla F_i(\theta_1)}_{T_2}\|.$$

Using triangle inequality, we obtain

$$\|\theta_1^+ - \theta_1^*\| \leq \|T_1\| + \|T_2\|,$$

and we control both the terms separately. Let us first focus on $\|T_1\|$.

**Bound** $\|T_1\|$   We first split $T_1$ in the following way:

$$T_1 = \underbrace{\theta_1 - \theta_1^* - \widehat{\gamma}\nabla F^1(\theta_1)}_{T_{11}} + \underbrace{\widehat{\gamma}\Big(\nabla F^1(\theta_1) - \frac{1}{|S_1 \cap S_1^*|}\sum_{i \in S_1 \cap S_1^*}\nabla F_i(\theta_1)\Big)}_{T_{12}}, \tag{18}$$

where $\widehat{\gamma} := \gamma \frac{|S_1 \cap S_1^*|}{m}$. Let us condition on $S_1$. According to standard analysis technique for gradient descent on strongly convex functions, we know that when $\widehat{\gamma} \leq \frac{1}{L}$,

$$\|T_{11}\| = \|\theta_1 - \theta_1^* - \widehat{\gamma}\nabla F^1(\theta_1)\| \leq (1 - \frac{\widehat{\gamma}\lambda L}{\lambda + L})\|\theta_1 - \theta_1^*\|. \tag{19}$$

Further, we have $\mathbb{E}[\|T_{12}\|^2] = \frac{v^2}{n'|S_1 \cap S_1^*|}$, which implies $\mathbb{E}[\|T_{12}\|] \leq \frac{v}{\sqrt{n'|S_1 \cap S_1^*|}}$, and thus by Markov's inequality, for any $\delta_0 > 0$, with probability at least $1 - \delta_0$,

$$\|T_{12}\| \leq \frac{v}{\delta_0 \sqrt{n'|S_1 \cap S_1^*|}}. \tag{20}$$

We then analyze $|S_1 \cap S_1^*|$. Similar to the proof of Theorem 1, we can show that $|S_1 \cap S_1^*|$ is large enough. From Lemma 3 and using our assumption, we see that the probability of correctly classifying any worker machine $i$, given by $\mathbb{P}(\mathcal{E}_i)$, satisfies $\mathbb{P}(\mathcal{E}_i) \geq \frac{1}{2}$. Recall $p = \min\{p_1, p_2, \ldots, p_k\}$, and we obtain $|S_1 \cap S_1^*| \geq \frac{1}{4}p_1 m$ with probability at least $1 - 2\exp(-cpm)$. Let us condition on $|S_1 \cap S_1^*| \geq \frac{1}{4}p_1 m$ and choose $\gamma = 1/L$. Then $\widehat{\gamma} \leq 1/L$ is satisfied, and on the other hand $\widehat{\gamma} \geq \frac{p}{4L}$. Plug this fact in (19), we obtain

$$\|T_{11}\| \leq (1 - \frac{p\lambda}{8L})\|\theta_1 - \theta_1^*\|. \tag{21}$$

We then combine (20) and (21) and have with probability at least $1 - \delta_0 - 2\exp(-cpm)$,

$$\|T_1\| \leq (1 - \frac{p\lambda}{8L})\|\theta_1 - \theta_1^*\| + \frac{2v}{\delta_0 L \sqrt{pmn'}}. \tag{22}$$

**Bound** $\|T_2\|$   Let us define $T_{2j} := \sum_{S_1 \cap S_j^*}\nabla F_i(\theta_1)$, $j \geq 2$. We have $T_2 = \frac{\gamma}{m}\sum_{j=2}^{k}T_{2j}$. We condition on $S_1$ and first analyze $T_{2j}$. We have

$$T_{2j} = |S_1 \cap S_j^*|\nabla F^j(\theta_1) + \sum_{i \in S_1 \cap S_j^*}\big(\nabla F_i(\theta_1) - \nabla F^j(\theta_1)\big). \tag{23}$$

Due to the smoothness of $F^j(\theta)$, we know that

$$\|\nabla F^j(\theta_1)\| \leq L\|\theta_1 - \theta_j^*\| \leq 3L, \tag{24}$$

where we use the fact that $\|\theta_1 - \theta_j^*\| \leq \|\theta_j^*\| + \|\theta_1^*\| + \|\theta_1 - \theta_1^*\| \leq 1 + 1 + \frac{1}{4}\sqrt{\frac{\lambda}{L}}\Delta \leq 3$. In addition, we have

$$\mathbb{E}\left[\left\|\sum_{i \in S_1 \cap S_j^*}\nabla F_i(\theta_1) - \nabla F^j(\theta_1)\right\|^2\right] = |S_1 \cap S_j^*|\frac{v^2}{n'},$$

which implies

$$\mathbb{E}\left[\left\|\sum_{i \in S_1 \cap S_j^*}\nabla F_i(\theta_1) - \nabla F^j(\theta_1)\right\|\right] \leq \sqrt{|S_1 \cap S_j^*|}\frac{v}{\sqrt{n'}},$$

and then according to Markov's inequality, for any $\delta_1 \in (0, 1)$, with probability at least $1 - \delta_1$,

$$\left\|\sum_{i \in S_1 \cap S_j^*}\nabla F_i(\theta_1) - \nabla F^j(\theta_1)\right\| \leq \sqrt{|S_1 \cap S_j^*|}\frac{v}{\delta_1 \sqrt{n'}}. \tag{25}$$

Then, by combining (24) and (25), we know that with probability at least $1 - \delta_1$,

$$\|T_{2j}\| \le 3L|S_1 \cap S_j^*| + \sqrt{|S_1 \cap S_j^*|}\frac{v}{\delta_1\sqrt{n'}}. \tag{26}$$

By union bound, we know that with probability at least $1 - k\delta_1$, (26) applies to all $j \ne 1$. Then, we have with probability at least $1 - k\delta_1$,

$$\|T_2\| \le \frac{3\gamma L}{m}|S_1 \cap \overline{S_1^*}| + \frac{\gamma v\sqrt{k}}{\delta_1 m\sqrt{n'}}\sqrt{|S_1 \cap \overline{S_1^*}|}. \tag{27}$$

According to Lemma 3, we know that

$$\mathbb{E}[|S_1 \cap \overline{S_1^*}|] \le c_1\frac{\eta^2 m}{\lambda^2\Delta^4 n'}.$$

Then by Markov's inequality, we know that with probability at least $1 - \delta_2$,

$$|S_1 \cap \overline{S_1^*}| \le c_1\frac{\eta^2 m}{\delta_2\lambda^2\Delta^4 n'}. \tag{28}$$

Now we combine (27) with (28) and obtain with probability at least $1 - k\delta_1 - \delta_2$,

$$\|T_2\| \le c_1\frac{\eta^2}{\delta_2\lambda^2\Delta^4 n'} + c_2\frac{v\eta\sqrt{k}}{\delta_1\sqrt{\delta_2}\lambda L\Delta^2\sqrt{m}n'}. \tag{29}$$

Combining (22) and (29), we know that with probability at least $1 - \delta_0 - k\delta_1 - \delta_2 - 2\exp(-cpm)$,

$$\|\theta_1^+ - \theta_1^*\| \le (1 - \frac{p\lambda}{8L})\|\theta_1 - \theta_1^*\| + \frac{2v}{\delta_0 L\sqrt{pmn'}} + c_1\frac{\eta^2}{\delta_2\lambda^2\Delta^4 n'} + c_2\frac{v\eta\sqrt{k}}{\delta_1\sqrt{\delta_2}\lambda L\Delta^2\sqrt{m}n'}. \tag{30}$$

In the following, we let $\delta_3 := \delta_0 + k\delta_1 + \delta_2 + 2\exp(-cpm)$, and

$$\varepsilon_0 = \frac{2v}{\delta_0 L\sqrt{pmn'}} + c_1\frac{\eta^2}{\delta_2\lambda^2\Delta^4 n'} + c_2\frac{v\eta\sqrt{k}}{\delta_1\sqrt{\delta_2}\lambda L\Delta^2\sqrt{m}n'}.$$

Let us simplify this expression. We first choose $\delta \in (0, 1)$ as the failure probability of the entire algorithm. Then, we choose

$$\delta_0 = \frac{p\lambda\delta}{CkL\log(mn')}, \quad \delta_1 = \frac{p\lambda\delta}{Ck^2 L\log(mn')}, \quad \delta_2 = \frac{p\lambda\delta}{CkL\log(mn')},$$

for some constant $C > 0$ that is large enough. In addition, since we assume that $p \gtrsim \frac{\log(mn')}{m}$, we have $\exp(-cpm) \le 1/\text{poly}(mn') \lesssim \frac{p\lambda\delta}{kL\log(mn')}$. Consider all these facts, we obtain

$$\delta_3 = \frac{4p\lambda\delta}{CkL\log(mn')}, \tag{31}$$

$$\varepsilon_0 \lesssim \frac{vk\log(mn')}{p^{3/2}\lambda\delta\sqrt{m}n'} + \frac{\eta^2 Lk\log(mn')}{p\lambda^3\delta\Delta^4 n'} + \frac{v\eta k^3\sqrt{L}\log^{3/2}(mn')}{p^{3/2}\lambda^{5/2}\delta^{3/2}\Delta^2\sqrt{m}n'}. \tag{32}$$

In addition, by union bound, we know that with probability at least $1 - k\delta_3$, for all $j \in [k]$,

$$\|\theta_j^+ - \theta_j^*\| \le (1 - \frac{p\lambda}{8L})\|\theta_j - \theta_j^*\| + \varepsilon_0. \tag{33}$$

## B.2 Convergence of the algorithm

We now analyze the convergence of the entire algorithm. First, we can verify that as long as

$$\varepsilon_0 \le \frac{p}{32}(\frac{\lambda}{L})^{3/2}\Delta, \tag{34}$$

we can guarantee that $\|\theta_j^+ - \theta_j^*\| \le \frac{1}{4}\sqrt{\frac{\lambda}{L}}\Delta$. We can also verify that as long as there is

$$\Delta \ge \widetilde{\mathcal{O}}(\max\{(n')^{-1/5}, m^{-1/6}(n')^{-1/3}\}), \tag{35}$$

using the definition of $\varepsilon_0$ in (32), we know that (34) holds. Here, in the $\widetilde{\mathcal{O}}$ notation, we omit the logarithmic factors and quantities that does not depend on $m$ and $n'$. In this case, we can iteratively apply (33) for $T$ iterations and obtain that with probability at least $1 - kT\delta_3$,

$$\|\theta_j^{(T)} - \theta_j^*\| \leq (1 - \frac{p\lambda}{8L})^T \|\theta_j^{(0)} - \theta_j^*\| + \frac{8L}{p\lambda}\varepsilon_0.$$

Then, we know that when we choose

$$T = \frac{8L}{p\lambda} \log\left(\frac{p\lambda\Delta}{32\varepsilon_0 L}\right), \tag{36}$$

we have

$$(1 - \frac{p\lambda}{8L})^T \|\theta_j^{(0)} - \theta_j^*\| \leq \exp(-\frac{p\lambda}{8L}T)\frac{1}{4}\sqrt{\frac{\lambda}{L}}\Delta \leq \frac{8}{p}\sqrt{\frac{L}{\lambda}}\varepsilon_0,$$

which implies $\|\theta_j^{(T)} - \theta_j^*\| \leq \frac{16L}{p\lambda}\varepsilon_0$. Finally, we check the failure probability. The failure probability is

$$kT\delta_3 \leq \frac{8kL}{p\lambda} \log\left(\frac{p\lambda\Delta}{32\varepsilon_0 L}\right) \frac{4p\lambda\delta}{CkL\log(mn')} = \frac{32\delta}{C} \frac{\log(\frac{p\lambda\Delta}{32\varepsilon_0 L})}{\log(mn')} \leq \delta \frac{\log(\frac{1}{\varepsilon_0})}{\log((mn')^{C/32})}.$$

On the other hand, according to (32), we know that

$$\frac{1}{\varepsilon_0} \leq \widetilde{\mathcal{O}}(\max\{\sqrt{mn'}, n'\}),$$

then, as long as $C$ is large enough, we can guarantee that $(mn')^{C/32} > \frac{1}{\varepsilon_0}$, which implies that the failure probability is upper bounded by $\delta$. Our final error floor can be obtained by redefining

$$\varepsilon := \frac{16L}{p\lambda}\varepsilon_0.$$

## B.3 Proof of Lemma 3

Without loss of generality, we bound the probability of $\mathcal{E}_i^{1,j}$ for some $j \neq 1$. We know that

$$\mathcal{E}_i^{1,j} = \left\{ F_i(\theta_1; \widehat{Z}_i) \geq F_i(\theta_j; \widehat{Z}_i) \right\},$$

where $\widehat{Z}_i$ is the set of $n'$ data points that we use to estimate the cluster identity in this iteration. In the following, we use the shorthand notation $F_i(\theta) := F_i(\theta; \widehat{Z}_i)$. We have

$$\mathbb{P}(\mathcal{E}_i^{1,j}) \leq \mathbb{P}(F_i(\theta_1) > t) + \mathbb{P}(F_i(\theta_j) \leq t)$$

for all $t \geq 0$. We choose $t = \frac{F^1(\theta_1) + F^1(\theta_j)}{2}$. With this choice, we obtain

$$\mathbb{P}(F_i(\theta_1) > t) = \mathbb{P}\left(F_i(\theta_1) > \frac{F^1(\theta_1) + F^1(\theta_j)}{2}\right) \tag{37}$$

$$= \mathbb{P}\left(F_i(\theta_1) - F^1(\theta_1) > \frac{F^1(\theta_j) - F^1(\theta_1)}{2}\right). \tag{38}$$

Similarly, for the second term, we have

$$\mathbb{P}(F_i(\theta_j) \leq t) = \mathbb{P}\left(F_i(\theta_j) - F^1(\theta_j) \leq -\frac{F^1(\theta_j) - F^1(\theta_1)}{2}\right). \tag{39}$$

Based on our assumption, we know that $\|\theta_j - \theta_1\| \geq \Delta - \frac{1}{4}\sqrt{\frac{\lambda}{L}}\Delta \geq \frac{3}{4}\Delta$. According to the strong convexity of $F^1(\cdot)$,

$$F^1(\theta_j) \geq F^1(\theta_1^*) + \frac{\lambda}{2}\|\theta_j - \theta_1^*\|^2 \geq F^1(\theta_1^*) + \frac{9\lambda}{32}\Delta^2,$$

and according to the smoothness of $F^1(\cdot)$,

$$F^1(\theta_1) \leq F^1(\theta_1^*) + \frac{L}{2}\|\theta_1 - \theta_1^*\|^2 \leq F^1(\theta_1^*) + \frac{L}{2}\frac{\lambda}{16L}\Delta^2 = F^1(\theta_1^*) + \frac{\lambda}{32}\Delta^2.$$

Therefore, $F^1(\theta_j) - F^1(\theta_1) \geq \frac{\lambda}{4}\Delta^2$. Then, according to Chebyshev's inequality, we obtain that $\mathbb{P}(F_i(\theta_1) > t) \leq \frac{64\eta^2}{\lambda^2\Delta^4 n'}$ and that $\mathbb{P}(F_i(\theta_j) \leq t) \leq \frac{64\eta^2}{\lambda^2\Delta^4 n'}$, which complete the proof.