[Reviews · NeurIPS 2020]

Review 1

Summary and Contributions: The paper proposes to address non-iid local datasets in distributed machine learning by clustering models during the learning process. That is, local learners make iterative estimations on cluster attribution and then optimize corresponding global models. Thus, every step of synchronization will send k (number of clusters) models to each of the local learners and each local learner returns its cluster and the update for its model. The paper provides a convergence rate for linear models with squared loss, and one for strongly convex learning problems, as well as an empirical analysis on MNIST and CIFAR.

Strengths: - The authors address an important problem of personalized distributed learning, i.e., the case of distributed learning when local learners do not have iid. data and they need a model that will suit exactly their local distribution. - The algorithm is sound and well explained.

Weaknesses: - the theoretical assumptions on the initial parameters appear to be too strong (see additional feedback for details) - the empirical evaluation is not fully convincing, since it (i) does not compare the approach to existing methods, but only with very simple baselines, and (ii) uses an unjustified success criterion (having the difference between right model and found solutions less than 0.6 of the noise deviation). Again, see additional feedback for details.

Correctness: The proofs to theorem 1 and 2 seem to be correct. For the experiments, the parameters of the algorithm used (step size, number of gradient steps) are reported, but it is not documented how they have been chosen. Was a separate dataset for parameter evaluation used?

Clarity: The paper is well written and easy to follow, it is a pleasure to read.

Relation to Prior Work: The paper adequately discusses related work and clearly stated the difference in the setup they consider, including a comparison with a recent pre-print [25].

Reproducibility: Yes

Additional Feedback: Empirical Analysis: - The approach is not compared to related work. Straight-forward baselines would be clustering on the central machine approach [9] or the fine-tuning of global models [7, 35] which are cited in the paper. - The criterion of counting the experiment as successful for the synthetic data is not truly justified: having the difference between right model and found solutions less than 0.6 of the noise deviation seems a bit arbitrary. - The approach to create clusters from MNIST and CIFAR by rotating images is interesting, but it seems weird that the most straight-forward approach (using subsets of the classes as clusters) was not evaluated. - It is also not specified which data was used for cluster identity check in the empirical evaluation - was it all the local data or only a particular subset used for the training step? Theoretical Analysis: My main concern with the theoretical analysis is the assumption that initial models are already very close their correct clusters (1/4 of the minimum distance between cluster centers for the linear models - for the strong convex problems an additional factor comes in that depends on the strong convexity and smoothness of the loss). I would argue that if models would be initialized this way, then performing a clustering on the initial models should already give the right clusters. A minor issue is that the convergence rate seems not to address the number of participating workers (line 4 of Algo. 1). It would be ok if the proof only applies if all workers participate, but this should be discussed. Additional comments: - similar to many other clustering techniques, the proposed approach requires knowing the right amount of clusters apriori. It would be great to see experiments on how the clustering behaves if the set number of clusters deviates from the true number of clusters. - since clusters reflect differences in local data distribution, the advantage of the proposed method over simple federated averaging should depend on the cluster dissimilarity, i.e., the difference in local data distributions. At which point do local distributions become so similar that it is favorable to use all local models in the aggregation, rather than clustering? At the same time, could there ever be a practical case with too many clusters? So far, the experiments only covered very small numbers of clusters (2-4). Overall, the proposed approach is fairly straight-forward, which is not necessarily a bad thing. The theoretical analysis is overall sound, but due to the strong assumption on the model initialization it is not fully convincing. My largest concern, however, is the empirical evaluation: since the approach is not compared to reasonable baselines, the success criteria are not clearly justified, and the way to produce clusters is a bit arbitrary, the experiments did not fully convince me that the approach is practically useful. I do believe that the proposed approach has potential and thus I encourage the authors to continue their research. However, in its current state, the paper seems to be not ready for publication. ----------------- After Author Response ------------------------------- I want to thank the authors for their detailed response and clarifying a few issues. I am still not fully convinced that the contribution is significant, but I am more positive about the manuscript. Consequently, I increase my score by one. On a side note: it is clear that you have chosen the hyperparameters that produced the best results, my question what precisely you mean by that: best results on a separate parameter tuning set, on the training set or on the test set?


Review 2

Summary and Contributions: This paper proposes a framework for clustered federated learning which can deal with the heterogeneity in the data distribution, especially the non-i.i.d. data. As for it's contribution, theoretical analysis of this work makes contributions to statistical estimation problems with latent variables in distributed settings

Strengths: It is proved that under the assumption of good initialization, the linear models and general strongly convex losses can obtain near the optimal statistical error rates. It is also showing that the IFCV is efficient in non-convex problems such as neural networks and outperforms the baselines on several clustered FL benchmarks created based on the MNIST and CIFAR-10 datasets by 5 - 8%. (1)Novelty: personalized predictions similarity among the users; consider a statistical setting with cluster structure (2)Significance: efficient in non-convex problem (3)Soundness: algorithm can succeed even if we relax the requirements on initialization with random initialization and multiple restarts.

Weaknesses: It doesn’t say this framework whether will work in non-linear problem. Moreover, the meaning of the global model and the local model didn’t explain very clearly. And also it may cause a privacy issue in identity information collection. The empirical evaluation lacks of comparisons with SOTA model.

Correctness: There is something wrong when I run the code, but in this article, the authors have clearly explained his thought and algorithm,and show impressive results.

Clarity: This paper is well organized and provides sufficient theoretical proof, however, a conclusion section is necessary in my opinion.

Relation to Prior Work: Yes

Reproducibility: Yes

Additional Feedback: I want to thank the authors for their detailed response and I will remain my score.


Review 3

Summary and Contributions: This paper investigates the problem of heterogeneity in federated learning with the clustered federated learning framework. A new algorithm, Iterative Federated Clustering Algorithm (IFCA), is introduced, in which the workers first estimate the cluster they belong to and then compute the corresponding updates. Convergence bounds are provided in two cases: linear Gaussian regression and strongly convex loss functions. Parametric experiments on synthetic show that the algorithm is able to retrieve approximate parameters in the linear case. Experiments on real data (modified MNIST & CIFAR10) show that the proposed IFCA yields better results than global or local training.

Strengths: - This paper introduces a novel algorithm for the previously introduced clustered FL framework, which is analysed both theoretically and empirically. I think this is one of the first analyses of such an algorithm, which is very significant. This makes it very relevant to the NeurIPS community. - As a byproduct, the theoretical analysis yields novel results for the statistical estimation of distributed models with latent variables.

Weaknesses: - The theoretical analysis is restricted to either a linear Gaussian case in relatively small dimension or to a strongly convex problem, with quite restrictive assumptions on the initialisation. - Experiments do not compare the results of the proposed algorithm to previous clustered federated learning algorithms (notably ClusteredFL, Sattler et al. 2019), making it difficult to appreciate the practical improvement. - Contrary to previous works (Sattler et al. 2019), this algorithm assumes that the number of clusters is known, which reduces its practical applications.

Correctness: Regarding the experiments, a comparison to ClusteredFL would have strengthened the paper. Besides that, the methodology seems sound.

Clarity: The paper is well written and easy to follow. However, a conclusion or discussion is missing.

Relation to Prior Work: The paper discusses well how it differs from previous work. However, it lacks experimental validation with respect to previous works.

Reproducibility: Yes

Additional Feedback: - Contrary to Sattler et al., the proposed algorithm is not communication-efficient as all models need to be broadcast to all clients participating in a round, which multiplies by K the amount of models to send. How would it be possible to reduce this communication cost? - Intuitively, the size of the mini batch on which the losses are evaluated in order to estimate the local cluster plays an important role, as it adds noise which prevents from finding the correct cluster ID. Did you always use the full dataset in the experiments? If not, did you see an effect of this batch size on the convergence speed?


Review 4

Summary and Contributions: Authors consider the setting of Federated Learning which each device belongs to a cluster, and data points within the same cluster follow the same model; it's a mixture of (linear) regressions model, where we have an additional information about subgroups which share the same latent variable. The model seems a bit simplistic to reflect the heterogeneity of data in the real world, but it is still a theoretical progress from an i.i.d. model. Authors prove the convergence rate of their algorithm to true parameters for both linear models and strongly convex loss functions. Experiments on synthetic data validates their theoretical findings, and experiments on semi-artificial data (rotated MNIST) shows the proposed method shall be applied to non-convex problems.

Strengths: The proposed algorithm is simple and thus allows a sound theoretical analysis, while still being applicable to models which don't allow theoretical analysis (such as neural networks). The proposed model has close connections to fundamental models in statistics, and theoretical results may have implications to audience broader than the Federated Learning community. I haven't carefully checked proofs, but results in this paper seem to align with known results.

Weaknesses: The proposed model seems to be too simplistic to reflect the heterogeneity of devices in the real world. At the least, large number of clusters $k$ would be expected in the real world; large $k$ will lead to small $p$, so the algorithm wouldn't scale very well with $k$. Rotated MNIST is an interesting idea to demonstrate the applicability of the proposed algorithm to non-convex problems, but it is still somewhat artificial as there is a small number of latent clusters, and doesn't convincingly demonstrate the practical usefulness of the algorithm. Authors did not address my review in their rebuttal, so I stayed with my original overall score.

Correctness: I wasn't able to closely verify proofs in the Appendix, but authors' claims are sensible, and are aligned with known results. Empirical evaluation is based on an artificial data, but this is reasonable for a theory paper.

Clarity: The paper is very well written. The outline of the proof clearly explains high-level steps needed for the final result, and each step is stated with a well-defined lemma and self-contained proof. Authors also compare their results with special cases of known results, which is a great sanity check.

Relation to Prior Work: Authors acknowledge that Mansour et al (2020) simultaneously proposed the same statistical model. Authors also make good connections to related work on Federated Learning for non-i.i.d. data and more classical models such as mixture of regressions model.

Reproducibility: Yes

Additional Feedback:

[Author Response · NeurIPS 2020]

We thank the reviewers for their comments. Below, we first respond to several common questions and then respond to
more specific questions raised by each individual reviewer.
**Common**
**Theory** All reviewers agree that our theoretical results are solid and well-explained. The only concern (from **R1, R3**)
is about our initialization condition. As mentioned in the paper (line 190-192), good initialization is a very standard
assumption in the convergence analysis of mixture models (such as clustering; see ref [1, 44]), due to the non-convex
optimization landscape of mixture model problems. In fact, in a paper by Jin et al 2016 titled *Local Maxima in the*
*Likelihood of Gaussian Mixture Models: Structural Results and Algorithmic Consequences*, it has been shown that bad
local minima provably exists in EM algorithms for Gaussian mixtures. As mentioned in line 58-60, in practice, we *do*
*not* need this assumption as random initialization with restarts works well. **R1** mentioned that with good initialization,
*"performing a clustering on the initial models should already give the right clusters"*. We argue that this does not hold in
practice: we usually observe that the number of wrongly clustered worker machines is high at the beginning of the
algorithm, and keeps decreasing as we run more iterations.
**Experiments** We observe that the following common questions about experiments were raised by the reviewers.
1) Comparison with other baseline algorithms and more realistic datasets: We conducted comparison with the one-shot
clustering algorithm proposed in ref [9]. In Table 1, we present the results on the Federated EMNIST (FEMNIST)
dataset which is one of the *realistic* federated learning datasets in the literature (see the paper by Caldas et al. 2018,
*LEAF: A Benchmark for Federated Settings*, i.e., ref [2] in our submission). The comparison on other datasets will be
added to the revised version.

Table 1: Test accuracy on FEMNIST

| IFCA ($k = 2$) | IFCA ($k = 3$) | one-shot ($k = 2$) | one-shot ($k = 3$) | global | local |
|---|---|---|---|---|---|
| 86.88 | 86.90 | 86.55 | 86.64 | 83.22 | 73.86 |

In this experiments, for IFCA and one-shot clustering algorithm, we share the representation layers among all the
models, but the last layers for different models are trained based on clustering. As we can see, the results of IFCA
are on par with the one-shot clustering algorithm. However, an important goal of FL is to reduce the computational
cost at the central server and take full advantage of on-device intelligence. **In the one-shot clustering algorithm,**
**the clustering is done at the center machine, which may lead to much higher computational cost at the center**
**compared to our algorithm.**
2) Knowledge of the number of clusters: Similar to many other clustering algorithms, our algorithm requires a hyper
parameter on the number of clusters. In our experiments, we observe that our algorithm is robust to the choice of
number of clusters. For example, on Rotated MNIST/CIFAR, when we choose the number of clusters larger than the
actual number, the algorithm can quickly identify that one of the cluster contains zero worker machines, and thus the
model can be discarded, when we choose the number of clusters smaller than the actual number, several clusters will
be classified as a single cluster, and in this case we can still improve over the *global model* and *local model* baselines.
Moreover, as we can see in Table 1, for FEMNIST, in which the clusters are more ambiguous, we also observe that our
algorithm is robust to the choice of number of clusters. We will provide more detailed discussions on the number of
clusters in the revision.
**Specific**
**R1** *"Was a separate dataset for parameter evaluation used?"*: We choose the hyperparameters within a wide range.
For each algorithm, we choose the hyperparmeters that produce the best result. This is a common method when running
experiments on public datasets.
**R1** *"criterion of counting the experiment as successful for the synthetic data is not truly justified"* The success criterion
in the synthetic data experiments only needs to be a constant multiple of the standard deviation of the noise. Our results
are robust to the choice of the constant, and we will clarify in the revision.
**R1** *"convergence rate seems not to address the number of participating workers"* As mentioned in line 154, we present
the results for full participation in order to streamline the analysis. Extensions to partial participation is straightforward.
**R2** *"whether ... will work in non-linear problem."* It will work for non-linear problems: we prove theoretical results for
strongly convex loss, which can be non-linear, and we show experimental results for neural networks.
**R2** *"privacy issue"* We did not aim to tackle it in this paper, but privacy is an interesting and important future direction.
**R3** *"comparison with ClusteredFL, Sattler et al. 2019"* We will add this comparison in the revision. We emphasize that
one important contribution of this work is the rigorous analysis of convergence rates, which Sattler et al. did not fully
address. In addition, in our algorithm, the worker machines identify their cluster membership by themselves, whereas
in Sattler et al., the clustering was done in a centralized manner, similar to ref [9]. Thus, our algorithm reduces the
computational cost of the central server, which is one of the major goals of FL as mentioned above.
**R3** *"reduce communication cost"* There are two ways to reduce communication cost: 1) when the cluster identity is
stable (which usually happens after running a few iterations), we don't need to send all the models to all the worker
machines, and instead we send the model corresponding to the worker's cluster, and 2) we can use weight sharing as
mentioned above (e.g., we only need to train $k$ different last layers). We will discuss these points in the revision.
**R3** *"mini batch ... to estimate the local cluster"* We use the full local data, and will clarify in the revision.

[Meta-Review · NeurIPS 2020]

Reviewers agree that the central idea is simple, which can be seen as a strength, and that the analysis is valuable. The concern about comparison only to baselines and not a more real-world method will be rectified by including the promised comparison to ClusteredFL. Without this comparison at submission, we must assume it will be on par, and therefore the significance of the result is reduced. The statements about reduced computation at the central server can also be accompanied by the statements abour privacy benefits (not sending user data to the server), even given the provisos at line 347.